# Brain Strategy Algorithm for Multiple Object Tracking Based on Merging Semantic Attributes and Appearance Features

**DOI:** 10.3390/s21227604

**Published:** 2021-11-16

**Authors:** Mai S. Diab, Mostafa A. Elhosseini, Mohamed S. El-Sayed, Hesham A. Ali

**Affiliations:** 1Faculty of Computer & Artificial Intelligence, Benha University, Benha 13511, Egypt; ms4elsayed@fci.bu.edu.eg; 2Intoolab Ltd., London WC2H 9JQ, UK; 3Computers Engineering and Control System, Faculty of Engineering, Mansoura University, Mansoura 35516, Egypt; melhosseini@mans.edu.eg (M.A.E.); h_arafat_ali@mans.edu.eg (H.A.A.); 4College of Computer Science and Engineering in Yanbu, Taibah University, Madinah 46421, Saudi Arabia; 5Faculty of Artificial Intelligence, Delta University for Science and Technology, Mansoura 35511, Egypt

**Keywords:** multiple object tracking, data association, dataset, deep learning, semantic attribute

## Abstract

The human brain can effortlessly perform vision processes using the visual system, which helps solve multi-object tracking (MOT) problems. However, few algorithms simulate human strategies for solving MOT. Therefore, devising a method that simulates human activity in vision has become a good choice for improving MOT results, especially occlusion. Eight brain strategies have been studied from a cognitive perspective and imitated to build a novel algorithm. Two of these strategies gave our algorithm novel and outstanding results, rescuing saccades and stimulus attributes. First, rescue saccades were imitated by detecting the occlusion state in each frame, representing the critical situation that the human brain saccades toward. Then, stimulus attributes were mimicked by using semantic attributes to reidentify the person in these occlusion states. Our algorithm favourably performs on the MOT17 dataset compared to state-of-the-art trackers. In addition, we created a new dataset of 40,000 images, 190,000 annotations and 4 classes to train the detection model to detect occlusion and semantic attributes. The experimental results demonstrate that our new dataset achieves an outstanding performance on the scaled YOLOv4 detection model by achieving a 0.89 mAP 0.5.

## 1. Introduction

Object tracking is of great interest to researchers because of its numerous computer vision applications such as robot navigation, self-driving, and smart surveillance. It attempts to give a unique ID for every object throughout all frames. However, many fundamental challenges abound regarding the development of a robust object tracking model. These include unpredictable changes in the background and the appearance of objects, full or partial occlusion, and data association. Over the years, researchers have developed robust algorithms to address these challenges. Their methods, however, have not solved one of the main problems in MOT, which is occlusion. Moreover, their algorithms have not reached human brain performance because not many have studied how the human brain deals with such challenges.

The human brain can identify both stationary and dynamic objects. Moreover, it can effortlessly perform vision processes, including activity recognition, image compression, and motion analysis. Therefore, we believe that the uncertainty in the tracking problem could be solved by developing an object tracking algorithm that can mimic the human brain. Merging researchers’ efforts in the neuroscientific community who have studied the approaches used by the human brain to track objects in real-time and researchers’ attempts in the computer vision community are needed to improve MOT models. This is what we have achieved in this paper.

In neuropsychology, the cognitive function of recognising people includes simultaneous routes of information and multiple processing stages. Models of these multiple processing stages have been provided from neuropsychological data obtained from case reports. According to research on young adults [1], humans can simultaneously recognise objects and extract meaning from what they see in the first half-second—but what part of the brain is responsible for this? Researchers in neuroscience have used advanced techniques to identify the brain areas that are involved in object tracking, for instance, magnetoencephalography (MEG), whole-brain functional magnetic resonance imaging (fMRI), and electrocorticography (ECoG) [2,3]. It has been found that this ability involves continuous activity within the anteromedial temporal cortex through the ventral temporal cortex [3]. While visual input is processed through this pathway, it has been found that it is transformed into an initial semantic representation (plants, animals) [4]. This happens in the inferior temporal cortex, before specific semantic representation emerges (rose, cat) in the anteromedial temporal cortex. Another conclusion drawn by [4] is that people recognition includes many attributes, such as the perception of faces, whole bodies, emotional expressions, clothes, gait, and individual marks. In other words, with these biological processors, humans can transform object representations or visual inputs into a format that is easily understood and extractable, regardless of the viewing conditions; this is the semantic attribute. In our algorithm, we aimed to design a tracker that mimics the working principle of the brain to enhance the performance of computer vision state-of-the-art algorithms. Our tracker is motivated by contemporary cognitive psychology. It combines an appearance feature vector (inspired from computer vision algorithms) and a semantic attribute (inspired by cognitive psychology) to represent the object. The appearance feature vector is used to track targets with no occlusion, while the semantic attributes are used to prevent incorrect tracking of an occluded object.

This research proposes novel brain-based MOT algorithms based on eight human brain concepts. Our approach, which resolves the frame-by-frame association problem, is novel; it combines appearance feature vectors and several semantic attributes (trousers, shirts, men, and women) to recognise objects. The main contributions are three-fold:We introduced a new dataset, PGC (Pedestrians’ Gender and Clothes Attributes), including a total of 40,000 images with more than 190,000 annotations for four classes (trousers, shirts, men, and women). We intend to ensure that the datasets are available and accessible to the public so that other researchers can benefit from our dataset and continue from where we ended. The reasons behind introducing our dataset as a main contribution are explained in Section 3.After evaluating our dataset by comparing it with an open dataset with the same classes using the same detection model, the results show that our PGC dataset facilitates the learning of robust class detectors with much a better generalisation performance.We introduced and evaluated a novel MOT algorithm that mimics the human brain, addresses occlusion problems, and improves the state-of-the-art performance on standard benchmark datasets.

The remainder of this paper is organised as follows: the next section explores various previous brain-inspired MOT algorithms. Section 3 presents the proposed method in detail. A quantitative evaluation for our algorithm (Merging Semantic Attributes and Appearance Feature MSA-AF), compared to five state-of-the-art algorithms, has been introduced in Section 4. Section 5 shows our algorithm’s limitations, while Section 6 presents what went wrong in our dataset. Section 7 concludes.

## 2. Related Work

Multiple object tracking has long been a widespread topic in computer vision, with a large number of publications. Over the last few years, researchers have put forward several brain-inspired tracking algorithms. However, these algorithms can only adopt one of the key cognitive effects used by the human brain, such as memory and attention, instead of imitating the entire process [5,6]. Other algorithms have integrated powerful tools that mimic the human nervous system to tackle the occlusion problem, such as deep neural networks (DNNs) [7], artificial neural networks (ANNs) [8], recurrent neural networks (RNNs) [9], and convolutional neural networks (CNNs) [10]. These networks can simulate the processes used by the human brain to identify, store, or process information. However, according to the research findings of [11], ANNs are fundamentally different from biological neural networks (BNNs) in many ways. For instance, cortical neurons in BNN and artificial neurons in ANN communicate to each other differently. BNN’s neurons are organised in a goal-driven manner, unlike ANN’s neurons. Additionally, training is not needed in BNNs, unlike ANNs, in which the weight of the neural connections needs to be trained to obtain better results.

The question that emerges here is how can we judge the degree to which a model agrees with biological outcomes? Despite no definite answer to this question in the literature, we will try to find answers using findings from the cognitive neuroscience of MOT and what these computer vision trackers have introduced in their models.

Researchers in cognitive science and neuroscience have executed many studies to ex- amine the cognitive mechanism behind MOT. According to their outcomes, MOT tasks focus on visual attention in early cognitive processing, while in later cognitive processing, visual short-term and working memory have been used [12]. Thus, we categorised the computer vision algorithms that imitate brain functions in their designs into two categories: memory-based algorithms and attention-based algorithms. We will cite a few MOT algorithms based on visual attention and short and long memory without being exhaustive.

Visual attention helps humans analyse complex scenes promptly and dedicate their limited cognitive and perceptual resources to the most relevant subsets of sensory data. Some human brain-inspired algorithms that achieved high accuracy predicted what part humans will attend to in an unobserved image. One of the best models for predicting fixation over images and videos is the decision theoretical model [13], which has shown promising results on the PASCAL 2006 dataset using the discriminant saliency model for visual recognition. This model has explained basic behavioural data that other models have explored less. In computer vision, attention is a convenient method for recognising targets by discriminating them from each other and the background. In [14], dual matching attention networks (DMAN) with a temporal and spatial attention mechanism strategy is proposed to allow the model to focus on matching similarity between each part of a pair of images.

The temporal attention network aims to investigate different observations in the trajectory by allocating different degrees of attention to them. The spatial attention module allows the network to focus on the matching patterns. Unlike [14], in which their attention model relies on only two aspects, [15] the temporal attention network focuses on four attention aspects to track the target. Two of these attention aspects are common with [14], which are temporal and spatial attention. The other two are layer-wise and channel-wise attention. However, the limitation of [15] is that their algorithm needs offline training to support feature selection in online tracking. Even though attention helps to tackle challenges such as background variability and variable target appearance, it suffers from occlusion due to its adaptability to variation. When a target is gradually occluded by an occluder, the algorithm misses the target because it tends to acknowledge it as a change in target appearance. Memory could be used to handle the occlusion by retaining the appearance information from the targets. [16] implement their network with long short-term memory (LSTM) units that improve their algorithm’s performance.

Visual memory is an essential cognitive mechanism that the human brain uses to deal with partial and full occlusion and to overcome variable target appearances. Makovski and Jiang [17] proved in their studies that the reason behind the observer’s recovery from any errors during tracking was the storing of surface properties inside the visual working memory. In [12], they propose a general theoretical framework for viewing human memory. They divide the memory model into three stages: the sensory register, the short-term store, and the long-term store. Many researchers in computer vision have used this theory to introduce brain-memory-inspired MOT [18,19]. In [18], they established a memory model to handle the main challenges in MOT by using the human brain three-stage memory theory [12]. Using this model, they proposed a template updating modelling algorithm that can remember and forget the target’s appearance. Their experimental results are promising for tackling sudden changes in appearance and occlusion. In [19], they are inspired by the same three-stage memory model. However, the three-stage memory model in this work aimed to find the right match between the appearance and the object by integrating the model into a multiagent coevolutionary process. Each agent can forget and remember the object’s appearance via its own experience through its memory system. Other researchers have used different strategies employing long short-term memory (LSTM) instead, such as [20,21,22]. Ref [20], introduced an algorithm based on deep reinforcement learning followed by the LSTM unit. Their track-by-detection algorithm detects the object and then applies an association model to solve the tracking problem. LSTM is used in the data association stage to address changes in appearance and occlusion. The experimental results of [20] show that their tracker is successful in handling scale changes, occlusion, and appearance similarity most of the time. However, their tracker faced some failures in some cases. For example, when the brightness of the environment changes, the detector fails to detect the object, leading to the tracker’s failure.

All algorithms introduced in this section are based only on attention or memory modules, added to their algorithm to imitate the human visual role. Although attention is undoubtedly involved in feature extraction and memory is involved in retaining extracted information, they are rarely used together in the same algorithm. In MSA-AF, we used not only both modules, but also six other strategies inspired by the human brain. We introduced the advantages and limitations of both models, memory and attention, in Table 1.

## 3. Proposed Algorithm

In this section, our tracking algorithm has been presented, which uses the outcomes from the neuroscience researchers’ community to imitate the human brain. What is critical in our work is to translate these biological findings into a computer vision algorithm. Both the neuroscience and computer vision communities share a general framework that focuses on the importance of memory, attention, and motion prediction to handle the MOT problem. To the best of our knowledge, no algorithms use all eight of the human brain strategies introduced in Table 2 as their building blocks. In contrast, our work was inspired by eight main findings from neuroscience research as building blocks to build MSA-AF. Table 2 shows these modules and how they inspired our algorithm. Integrating the eight modules together is undoubtedly ambitious, leading to improvements in the state-of-the-art MOT algorithms. However, strategies 4 and 8 from Table 2 played the most critical role in the final association stage, as discussed in more detail later on in this section.

The basic flow chart of the tracking algorithm we implemented in this work is illustrated in Figure 1. Our focus was on handling the existing challenges facing state-of-the-art algorithms, the most important of which was occlusion. The first building block in our algorithm is detection, which takes a frame and gives us the bounding boxes (Bbox) of the four classes (man, woman, shirt, trouser) inside the image. Then, the second block will take the Bbox from the previous frame, apply the Kalman filter [26] to predict their new position in the current frame, and then apply the cosine metric learning method for learning a feature space to associate the detection with the objects. The last three building blocks distinguish MSA-AF from any other MOT in the field of computer vision, which is similar to how human logic works to solve a tracking problem. This occurred by transferring the information in the image into semantic information that is saved in the memory for retrieval when it is needed. This brings us to the last building block, which uses the saved semantic information to decide whether occlusion is happening or not and how to deal with it. The rest of this section will introduce the logic behind each building block and how all are connected to imitate the human brain and improve the performance of recent MOT algorithms.

### 3.1. Detection

Without exaggerating, we can say that the detection step is essential for any tracking-by-detection algorithm such as ours. Missing, false, or non-accurate detection will lead to poor MOT performance; consequently, more time is spent on this stage to secure a good foundation for our algorithm. Some algorithms assume that they have the detections needed to focus on the tracking step, such as in [24]. Three crucial questions need to be answered to deal with the detection step successfully: What classes do we need to detect? What is the detection model that will be used? Do we have a good dataset for the training? All these questions will be answered in this section. Figure 2 depicts the different phases used to ensure a reliable output from the detection stage.

#### 3.1.1. Classes

In any MOT algorithm that deals with people, the person is the class that needs to be detected. In MSA-AF, as we imitate the human brain, we use a different approach. Biological evidence shows that the human brain transforms the visual input into an initial and then a specific semantic attribute [4]. We employ the person reidentification approach used in multi-camera networks to find a semantic attribute representing a person. This approach identifies the best attributes for recognising a person and helps an operator to track, search, and locate the target object. For example, in the Van Koppen and Lochun [29] survey, over 1313 human descriptions were collected from individuals who witnessed a robbery incident. They found that of the 43 categories identified, only 30% of the eyewitnesses accurately described all nine categories. The categories included skin colour, hair type, height, gender, appearance (which includes race), accent, hair colour, build, and age.

Similarly, Sporer [30] examined 139 descriptions provided by 100 witnesses. He discovered that of all the descriptions, 31% mentioned clothing information, 29.6% detailed facial features, 22% contained information on movement features and physical attributes (height, race, and age), and 5% included personality information. After considering the findings of these two studies, we selected clothes (trousers and shirt) and gender (man and women) as our semantic attributes. We will reserve the term semantic information units to refer to the trouser, shirt, man, woman semantic attributes.

#### 3.1.2. Dataset

The training dataset can be defined as the initial data used to develop the machine learning model. A high-quality training dataset, without a doubt, is the critical item for any machine learning system. Most performance detection models can be rendered useless without high-quality and quantity training datasets.

After deciding what classes we needed to detect, we searched for a dataset to train the detection model with the same classes. However, we could not find any dataset with shirt, trouser, man, and woman classes that could benefit our algorithm. Thus, we built the PGC dataset not only to serve our algorithm, but also to benefit other researchers.

Uniqueness: We argue that there is no existing state of the art dataset like the PGC dataset. Existing datasets either have the person as a single class, such as MSCOCO [31], or recognise pedestrian attributes, such as PETA [32]. MSCOCO, like most detection datasets, has many different classes (81)—one of them representing a person, without any classes representing any details about the person. In contrast, like all pedestrian attribute datasets, the PETA dataset works on a set of images that show only one person and annotate a set of binary attributes to this person. Lable (upperBodyBlack, lowerBodyGrey, hairBlack, footwearWhite, lowerBodyCasual, lowerBodyJeans, personalLess30, personalMale) is one example of annotation in the PETA dataset. As we can see, this gives information about the person rather than where these attributes are in the image. On the other hand, PGC has four classes that represent attributes used by the human brain to identify a person, with the location of each attribute inside the image. Add to that the uniqueness of the PGC in the process of acquiring, annotating, and testing that will be discussed later in this section.

Usage: The PGC dataset can serve many fields in computer vision. For instance, pedestrian reidentification, detection and tracking research. Moreover, PGC can be integrated with different types of clothes instead of only shirts and trousers to serve fashion research. High-quality training data is a daunting task, so the quality and quantity could be compromised to speed up the procedure. In contrast, we prioritise quality over speed throughout the collection the annotation stages of this study.

At the data collection level, diversity is the main factor that leads to high-quality data. Representation bias is one of 23 types of bias introduced by [33] that could lead to unfairness in machine learning. For instance, [34] remark that some datasets such as IJB-A [35], which compiled 79.6% lighter-skinned faces with just 4.4% of images from dark-skinned female faces, show bias. Algorithms that rely on these datasets suffer from bad performance, due to the representation bias in the training data. Therefore, we considered diversity and quantity when collecting 30,000 images from the web and 10,000 from the PETA [32] dataset. To ensure that the PGC dataset includes all scenarios in the real world, we searched for specific events and a specific group of people; dark-skinned, light-skinned, man, woman, Asian, man or woman with headcover, Arab, old, and young people are examples of a diverse group of people we searched to include in our dataset. Another critical factor in our search is the scene’s background, such as indoor, outdoor, night, or morning. The position of the person is a consistently underestimated factor in any pedestrian dataset. Therefore, it is essential for the training dataset to have all possible body positions such as standing, sitting, walking, running, side-view, back-view, front-view, high camera angle, entire body, and upper body part. Additionally, a wide range of image resolutions in any dataset makes the model robust to any movement in the camera or even to poor resolution images. Finally, we were eager to include images with occlusion as much as possible, which helped us handle the occlusion in the tracking step. Although applying a diversity search in the PGC dataset did slow the collection process, its outcome is outstanding, as shown in Section 4. The image collection in Figure 3 has been taken from the output of the scaled YOLOv4 detection model on our dataset. It is a random selection taken by the detection model for testing. As we can see, it shows the diversity of our dataset.

Data annotation is the most critical and time-consuming step in the detection process. Two main factors are needed to achieve high-quality annotations: annotation tools and rules to guide the annotation procedure. We have used the Roboflow [16] tool to annotate our dataset. Using Roboflow made the process of annotation easier and faster. While the annotation tools speed up the process, the annotation rules increases its quality. Appendix A shows the annotation process and regulations that have been applied to our dataset. The main rule that needs to be known is that shirt class does not mean only shirts, but any clothes on the upper body such as a coat, jacket, sweater, or T-shirt. Same with the trousers, which could be shorts, leggings, jeans. Although this could be a limitation to our dataset, it would be possible for any researcher to change the labels on the shirts annotation to any other clothes they aim to investigate in their work. Including a wide range of clothes in one class has benefited our tracking algorithm.

Regarding dataset size, the most common question for researchers is how many images are needed to train the model? There is no magical equation regarding the dataset size that could tell us how many images are needed to train a detection model. However, it is a common understanding that having more training data leads to better performance. We discovered how much training data we needed by building the detection model at first with 10,000 images and checked how it performed. Then we increased it by another 10,000 until we reached our target performance. We stopped collecting data after 40,000 images, as that is when the model performance reached our desired performance.

To sum up, in the field of computer vision, and especially object detection, a good quality dataset is the core of any successful model, which make a good quality dataset a contribution that aims to improve the model performance. However, the scope of the time needed to organize, annotate, and clean the dataset still tends to be underestimated. Five remarkable criteria that show how PGC dataset contributes to the state of the art are introduced next: (1) the PGC dataset is unique—no other dataset in object detection or object tracking works using pedestrian attributes such as man, woman (using all of the body, not only faces), shirt and trouser. All known datasets, such as MSCOCO and PASCAL, works on the whole person as a single entity. (2) Rich annotation: almost all images have at least three classes out of four. (3) Large size: not only the number of images matters but also the number of annotations per class; our dataset has 190,000 annotations in 40,000 images for only four classes. Figure 4 shows the number of annotations for each class. (4) High diversity: we spent a lot of time introducing deliberately selected images; Figure 3 is proof of our dataset diversity. (5) Balance between classes: Figure 4 shows how PGC dataset classes were balanced, in which men and women were represented equally, with 34,163 women and 35,252 men.

#### 3.1.3. Detection Model

Recent advances in methods for object detection are one of the main factors responsible for the success of MSA-AF. Tracking-by-detection algorithms such as ours rely on the quality of the detection model. Therefore, we searched for a detection model that achieves high performance at a reasonable speed. Deep neural network-based detection [36,37,38,39] showed a high performance compared to state-of-the-art detection algorithms. Region-based convolutional neural network (R-CNN)-based detection algorithms use the region to recognise the object inside the image. These algorithms do not look at all images, but only at the parts with a higher chance of containing the object. Unlike R-CNN-based detection algorithms, the YOLO family take the entire image in one run and predict the bounding boxes, as well as the class probability for these boxes. The YOLO family is commonly used for real-time detection, as it can process 45 frames per second.

This method was first described in 2015 [38], and many versions have been released since then, up until the latest version, published in November 2020 [40]. Not all versions of YOLO are published in a paper, such as YOLOv5; neither do they all have the same authors. Figure 5 summarises the performance and the timeline of all these versions; to conclude, we choose two of these versions for comparison, and finally choose just one of them. The results of our experiment in Section 4 between YOLOv5 and scaled YOLOv4 [41] shows that scaled YOLOv4 showed promising results for our dataset. Thus, scaled YOLOv4 was used as the detection model in MSA-AF.

### 3.2. First Association Stage

The main target of this stage is to decide whether the detection, which is the output from the scaled Yolov4, is associated with an existing track from previous frames or not. If the tracks did not associate with any detections, they would be added to the unassociated tracks list—the same with detections that did not associate with any existing tracks, which would be added to the unassociated detections list. While the first association stage deals with associated tracks and detection, both unassociated tracks and detections will be solved during the next stage—the final association stage.

Firstly, we need to predict the new position of the tracks in the current frame using the Kalman filter, and then build association cost metrics that will be used to solve the first stage of association between the detections and the predicted tracks. The output of this stage will guide the next step, where the final association decision will be made.

Prediction is an essential step that the human brain uses to track an object [25]. The Kalman filter has been used to predict the new position in many MOT algorithms due to its simplicity, versatility, and effectiveness. The complete mathematical derivations and formulation of the Kalman filter are beyond the scope of our paper. What we need to acknowledge is the dynamic model of target motion that the Kalman filter will use. The constant velocity model is suited for use in our framework as people hardly ever leave a constant velocity. In this model, we assume that the velocity is constant during the tracking so that we can predict the new position of the target easily using the Bbox location and the velocity information.

In the first association stage, the association cost metric has been used to decide which list from the three lists mentioned earlier the tracks and detections belong to. We know from cognitive study findings that the human brain uses all available information, such as appearance information, to discriminate an object [2]. Consequently, we decided to use appearance information to discriminate targets from each other rather than to discriminate them from backgrounds. CDA-DDAL [42] and DeepSort [24] learned the appearance features from the PETA reidentification task; we used the association metric from Deep sort [24]. Wojke et al. [24] used the person reidentification task to increase discrimination by applying the deep feature extractor from a wide residual network (WRN). The primary motive behind using the deep appearance description from Wojk et al. is that their network obtained a competitive performance for online tracks in real-time speed. Although appearance information can represent and discriminate the objects effectively, alone it is not enough. Adding spatial information is essential for forming a robust association cost metric. The Mahalanobis distance (1) between the predicted Kalman state of the tracks and the detection measure *KD*(*i*,*j*) will represent the spatial information, while the cosine distance (2) between the appearance information of the tracks and the detections *AD*(*i*,*j*) will represent the appearance information. Equation (3) shows this combination, which we believe would be representative enough at this stage.
(1)$$KD\left(i,j\right)=\frac{{\left(d_{j}-\mu_{i}\right)}^{T}}{S_{i}\left(d_{j}-\mu_{i}\right)}$$
where KD(i,j) represent the Mahalanobis distance of detections Bbox $d_{j}$ from *i*th tracks with mean $\mu_{i}$. The covariance matrix is *S*.
(2)$$AD\left(i,j\right)=\min\left\{1-r_{j}^{T}r_{k}^{\left(i\right)}|r_{k}^{\left(i\right)}\in R_{i}\right\}$$
where AD(i,j) is the smallest cosine distance between the *j*th detections and *i*th tracks, and $r_{j}^{T}$ represents the appearance distributor of detections $d_{j}$, while $r_{k}^{\left(i\right)}$ is the appearance descriptor of the last *k* associations, which is sets at 100 [24].
(3)$$A_{i,j}=\lambda KD\left(i,j\right)+\left(1-\lambda \right)AD\left(i,j\right)$$

$A_{i,j}$ represents the association cost metric which combines the appearance and spatial representation of the object, while λ allows us to control the impact of each metric on the overall cost metrics.

Solving the association problem by using the Hungarian algorithm on the association cost metrics is the last step in this stage, generating three lists—the first list includes all detections and tracks associated with each other. The second list will consist of the tracks that did not associate with any detections in the current frame. The third list includes the detections that did not match with any tracks. In [24], they deleted the tracks in the second list if they exceeded a certain age and created a new track for all detections in the third list. However, not all unassociated tracks exiting from the scene are deleted; they could be in a long-occlusion. Moreover, not all unassociated detections are new tracks; they could be long-term occluded tracks reappearing in the scene. This was the reason for introducing the last decision stage, which solved all these questions using the attributes.

### 3.3. Final Association Stage

Semantic attributes have been used in many detection algorithms [43]. However, they have never been used to solve occlusion in MOT algorithms, to the best of our knowledge. Additionally, semantic attributes play a significant role in the human brain by discriminating targets during tracking [3]. We used these semantic attributes in MSA-AF during the final association decision for those two reasons: not only to detect the presence of occlusion but also to solve the occlusion problem. E. Jefferies [44] raised an essential question regarding the organisation of semantic attributes: how is semantic information linked together in our brain to generate a unitary experience of a person? In neuropsychology, there are two principles regarding the organisation of semantic information: semantic hub and distributed only. In the semantic hub model, the access to knowledge occurs through a semantic hub, which causes an inability to link any semantic information if damage occurs to the central hub [45]. On the other hand, the distributed-only model suggests that different types of semantic information interact directly without central hub [46]. We combined both models to benefit our algorithm as follows: The central hub will be the person track, which will contain all semantic, appearance, and spatial information, with direct access to all other semantic attributes (man, woman, shirt, trouser). At the same time, direct interactions between semantic attributes are necessary in case of any disappearance of these attributes. Figure 6 shows the relationship between these attributes and the central hub. The direct interactions between the four semantic attributes are the key step in this stage. In other words, how can we decide which shirt belongs to which person? In a normal situation where no occlusion occurs, bounding box overlap could be enough to make that decision. However, in the occlusion situation, more than one shirt could overlap with the same person. For this reason, we introduced a metric called intersection over attribute (IOA); this attribute here could be a shirt or trouser. Most MOT algorithm use intersection over union (IOU), which only indicates the overlap between Bboxes but does not give any information about occlusion.
(4)$$\mathrm{IOA}\;=\frac{\mathrm{Area}\;\mathrm{of}\;\mathrm{Overlap}}{\mathrm{Area}\;\mathrm{of}\;\mathrm{Attribute}}$$

IOA not only tells us which attributes belong to which person but can also be used to identify if the person is in an occlusion state or not. The ratio of the overlap between a person and an attribute (shirt or trouser), and the attribute area IOA, would be close to one if all of the attribute’s bounding box is inside the person’s bounding box. In this case, the shirt will be linked to that person as their shirt (see Figure 7a). In the occlusion state, the IOA will be less than 0.9 and more than 0.4, meaning that the person overlapped with parts of these attributes. An example of the three stages of occlusion that define the final decision is shown in Figure 7.

The skeleton outline for using IOA in the final association decision is detailed in Algorithms 1–3. In the three algorithms, Tr refers to track IDs from the Associated_Man_list and the Associated_Woman_list, and Sh refers to the shirt IDs from the associated_shirt_list. At the same time, Sh.Track_bbox refers to the Bbox of the person that a shirt belongs to and the Tr.Shirt_bbox refers to the Bbox of the shirt belong to the person. Tr.Shirt_age counts how many frames the shirt of this person has been tracked in, Sh.Tracks_age counts how many frames the person linked to this shirt has been tracked in, and Sh.Occlusion_age counts how many frames the shirt was occluded in. Tr.Occlusion_status indicates if the person is in an occlusion situation or not, while Sh.Occlusion_status shows if the shirt is in an occlusion situation or not, and Sh.Occluded_with gives the ID of the person that occluded with that shirt. Finally, time_since_update gives the number of frames in which the person did not update.
**Algorithm 1** Associated tracks and detection list**Input:** List of associated tracks and detections**Output:** Detecting occlusion state and save semantic attribute1For Tr in Associated_Man _Woman_list2  For Sh in Associated_Shirt_list3     IOA = Intersection_over_ShirtArea(Tr,Sh)4       If IOA > 0.95           If Tr.Shirt_id = Shirt_id and Sh.Track_id = Track_id6             update Sh.Track_bbox & Tr.Shirt_bbox7             Tr.Shirt_age = + 18             Sh.Track_age = + 19           Else10             Tr.Occlusion_status = true11             Sh.Occlusion_status = true12             Sh.Occluded_with = Tr.id13       else14            If IOA > 0.415             Tr.Occlusion_status = true16             Sh.Occlusion_status = true17             Sh.Occluded_with = Tr.id

**Algorithm 2** Unassociated tracks list**Input:** List of unassociated tracks**Output:** Updated model
1For Tr in unmatched_Person2    If Tr.Occlusion_status == true3        Tr.Occlusion_age = + 14        time_since_update = + 15          If Tr.Shirt_id in matches_Shirt6            update Tr.Shirt_bbox7            Sh.Track_age = 08            Sh.Track_Occlusion_age = + 19            Shirt_age = + 110          Else11            Tr.Occlusion_age = + 112            time_since_update = + 113    Else14     time_since_update = + 1


**Algorithm 3** Unassociated detections list**Input:** List of unassociated detections**Output:** Retrieve the obj ID after appearing from occlusion
1For Det in unmatched_detection_person2For Sh in matches_Shirt3    IOS = Intersection_over_ShirtArea (Det,Sh)4      If IOS > 0.95            If Tr.Occlusion_age (Sh.Track_id) > 16                Tr.Occlusion_age (Sh.Track_id) = 07                  Tr.age = 18                  Track.update (Sh.Track_id.kf, Det)9                    Sh.Track.bbox = Det.bbox10                  Assosiated = True11            If Tr.time_since_updated > 012                    Tr.time_since_updated = 013                      Tr.age = 114                      Track.update (Sh.Track_id.kf, Det)15                        Sh.Track.bbox = Det.bbox16                        Assosiated = True17  If Assosiated = False18     initiate_track (Det (Det_idx],classes (Det_idx].item())


Each algorithm from the three has a different input list: associated tracks and detections list, unassociated tracks list, and unassociated detections list. We will work on each list separately to collect as much information as possible to feed to the main hub, which is the person, then take a suitable final decision depending on this information. The algorithm for each list will be repeated for the trousers too. As we can see from Algorithm 1, IOA has been calculated in line 3 to define whether a shirt belongs to that track or not, then this information was added to the person track. Moreover, the occlusion state will also be detected in line 9.

Algorithm 2 deals with tracks that disappeared in the current frame; this could be occlusion or a detection step problem. In Section 3, we introduced our scheme to ensure a high-quality output from the detection stage. However, what if the detection model fails to detect an object for a few frames or detects the object inaccurately? How will our algorithm solve such a problem? In this case, we need to know the reason for the disappearance to make the right decision. To that end, the information that we collected in Algorithm 1 was used, plus the age of the track, to decide whether we needed to forget this track, delete it, or keep it until it appears again. This step is what distinguishes MSA-AF. The knowledge of the occlusion state of the track will make us not rush to delete these tracks, even if the age exceeds the threshold age, as in line 3.

Even if the tracks are not in an occlusion state, they could disappear because of the detection problem. Therefore, we keep updating the tracks state for several frames to keep identifying the tracks when the detector model detects them again. We will keep track of the shirt and the trousers that belong to this person (line 6) if they are still detected, which is the case in Figure 7c.

Algorithm 3 includes the final decision concerning whether any unassociated detection is a new person, to create a new track; a person coming out of full or partial occlusion; or if that detection was a track missed due to a detection problem. Line 4 in Algorithm 3 finds the IOA between the unassociated person and the shirts in the associated detection, to see if they belong to that person or not.

## 4. Experiment Results

In this section, the results of our tracking algorithm and dataset will be introduced. Moreover, the comparison between the detection model we used and the nearest competitive model was presented, to prove that the model we used gives the best MT (mostly tracker metric).

### 4.1. Performance Metrics

Detection models are usually tested using precision, recall, and mAP at 0.5 or 0.5–0.95. Precision is the positive prediction value (of all the positive predictions, how many are true positive predictions). The recall metric is the true positive rate (of all the actual positives, how many are true positive predictions). These are then plotted to get a PR (precision-recall) curve; the area under the PR curve is called average precision (AP). Now we have AP per class; mAP is the averaged AP over all the object categories. Therefore, all detection models seek a high mAP value.

Regarding tracking performance metrics, defining the metrics that can say if MSA-AF solved the tracking problem in videos better than state-of-the-art algorithms is a challenge by itself. MOTChallenege [28] uses many metrics to evaluate the benchmark performance of MOT algorithms. We have used eight of these metrics to evaluate our algorithm. Multiple objects tracking accuracies (MOTA) [47] measure three errors: identity switches, missed targets, and false positives. In contrast, multiple objects tracking precision (MOTP) [47] measures the misalignment between the predicted Bbox and the annotated Bboxes. ID F1 [48] combines ID precision and ID recall by measuring the correctly identified and computed detection ratio and the ground truth average number. Ground truth trajectories covered by track trajectory for more than 80% of their life span were measured as the most tracked target (MT). In contrast, mostly lost (ML) measures the ground truth trajectory, covered for less than 20% of its life span by the track trajectory. Rcll is the ratio of correctly identified detections to the total number of ground truth boxes, and Prcn is the ratio of true positive TPs to the addition of true positives and false negatives. Finally, the FAF is the average number of false alarms per frame.

### 4.2. Detection Performance

Any detection algorithm’s output depends on two main factors: the dataset and the detection model. Here, we will take each factor and show how it affected the overall performance of our MOT algorithm.

#### 4.2.1. Dataset

The PGC dataset can be used in visual surveillance research on pedestrian detection, tracking, fashion projects, and reidentification. The key challenge of evaluating our dataset on the detection model is that we cannot know how it will perform until we test the model on new data that the model was not trained on. Therefore, we divided our dataset into a training set and a validation set. Then, the loss function for the two sets was evaluated, as shown in Figure 8; the smaller the loss, the better the classification performance.

Based on this diagram, it is clear that our model does not show any signs of overfitting. First, as the validation loss plot decreases, so does the training loss plot, while there is a small gap between the two. Second, there is no infliction point on the validation loss plot, which indicates that the training process did not receive sufficient experience with the data. Thus, the model still fits well.

In order to avoid overfitting, which resulted from the search for the perfect fit, the training ended at 300 epochs, based on early stopping.

Compared with the training set, the validation set had a smaller loss. Along with the early stopping criteria, YOLO uses a regularization approach to help regularize the model.

It could be a surprise that the loss of the validation set is less than that of the training set. The reason behind this is that the regularization loss did not add to the validation loss, but only to the training loss. YOLO uses a regularization method—batch normalization—after its convolutional layers to help regularize the model, which improves the mAP by 2%.

After proving that our dataset was not biased and did not overfit the model, we started to search how can we compare the PGC dataset to another dataset. Looking for how the most famous dataset in the field of object detection compared their datasets with others and imitating that was the best fit for our work. The extensive dataset, MSCOCO [31], has played an essential role in boosting object detection and tracking research—especially in person detection—by including more than 700,000 instances of the person category. Another dataset widely used in detection models is the Pascal VOC 2012 dataset, which has 17,118 images across 20 object classes with 13,168 person annotations [49]. Although MSCOCO has way more person instances in their dataset than PASCAL, PASCAL outperforms MSCOCO in 6 categories out of 20, including the person category. MSCOCO’s results confirm that the annotation number could not be the main or the only factor differentiating the dataset. MSCOCO compared their dataset performance to the PASCAL VOC 12 dataset performance by using both datasets to train the same detection model; then, they used the performance differences of the model across the two datasets for comparison. The way MSCOCO compares their dataset has been used to examine our dataset and demonstrate its high quality. We have used scaled YOLOv4 as the detection model for measuring PGC dataset performance, for reasons given in Section 3.

The challenge in this evaluation was finding a dataset with the same classes we used (man, woman, shirt, trouser). As we mentioned previously, all known datasets have worked on persons as single entities, without any semantic attributes. The only option we had for evaluating our dataset was to collect the same amount of images, 40,000, with the same classes from the Open Images DatasetV6 [50]. Then, we trained scaled YOLOv4 using both datasets and compared their performance. However, Images DatasetV6 had only 23,000 images with our four classes. Therefore, we compared 23,000 images from our dataset with 23,000 from Image DatasetV6 to have a fair comparison. Figure 9 shows how PGC dataset classes are more balanced than Open Images Dataset V6 classes. Figure 10 shows the success of our dataset in outperforming every single metric—even the training time, as discussed next. Using the clarification at the performance metrics subsection, the higher the precision, recall, and mAP, the higher the model’s performance. Our dataset gains higher precision, recall, and mAP. The mAP value of our dataset at 0.5 and 0.5–0.95 proves how rich, reliable, and robust our dataset is. The PGC dataset performs equally well in the four classes, unlike most big datasets that outperform in one class more than others; this is due to the balance. Figure 11 demonstrates the performance of the four classes in the YOLOv5 model.

Table 3 shows a comparison between the PGC dataset and the Open Images Dataset V6 across performance metrics, plus the training time. We used an intel core i7 computer with 64GB Ram and NIVIDA GeForce RTX2070 to train the detection models.

#### 4.2.2. Detection Model

Scaled YOLOv4, YOLOv5l, YOLOv5m, and YOLOv5s have been tested using the PGC dataset, in order to choose one of them for use in our MOT Algorithm. The reason behind using these four models is introduced in detail in Section 3. Although the training time of scaled YOLOv4 is much longer than the training time of the YOLOv5 family, the outstanding performance of scaled YOLOv4 on the PGC dataset is worth the long training time. While scaled YOLOv4 took 3 d 1 h 27 min 6 s to train 300 epochs, YLOLv5l took 1 h 23 min 12 s, YOLOv5m took 55 min 24 s, and YOLOv5s took 34 min 19 s. Figure 12 shows the performance of the four models using two metrics: mAP 0.5 and mAP 0.5–0.95. As a result of this comparison, scaled YOLOv4 with 0.8924 mAP was chosen as the detection model for MSA-AF.

### 4.3. Tracking Performance

We tested MSA-AF on MOT17 and MOT20 challenges. Each challenge had several videos with zoomed-in and zoomed-out scenes. However, all videos in the MOT20 dataset had a crowded scene, unlike MOT17.

To evaluate the performance of our algorithm, we compared our results with five other MOT algorithms. LPC_MOT [51] introduced a proposal-based MOT learnable framework, while MPNTrack [52] used deep learning for feature extraction and association. GNNMatch [53] used graph convolutional neural networks (GCNN) on top of convolutional-based features for object association. At the same time, Tracktor++v2 [54] takes advantage of a detector’s regression head to perform temporal realignment of the object’s Bbox. The concepts of SORT17 [55] have been used in MSA-AF, which use Kalman filtering in the image space and the Hungarian method on the association metric to solve the association problem. Table 4 illustrates the comparison between the performance of these five algorithms and ours on the MOT17 dataset. As shown in Table 4, our algorithm achieved state-of-the-art results, especially in MOTP, ML, and FAF. We attribute this performance to our dataset, which provides high-quality detections to the first and final association decision. Moreover, our final association decision model decreased the number of lost tracks, leading our algorithm to have the lowest ML.

On the other hand, Table 5 compares the same algorithms, using the same evaluation metrics but on a more challenging dataset, to MOT20. All MOT20 videos have a crowded scene, and the pedestrians are too far from the camera, as shown in Figure 13. Using the MOT20 dataset makes the job of our algorithm much harder, because it relies on detecting semantic attributes, which could be a challenge in most MOT20 videos. However, MSA-AF did surprisingly well in the MOT20 dataset. Although the performance of MSA-AF on the MOT20 dataset is not as good as on MOT17, it still has the best MT and ML. These results prove that the robustness of our final association stage is high.

It is worth mentioning here why we believe that MSA-AF is competitive with the results of the other algorithm. First, as shown in Table 4, MSA-AF dominated at MOTP, ML, and FAF metrics. Second, although the other five algorithms that we compared MSA-AF with from MOTChallenge use given detection as an input instead of detecting the object, it exceeds SORT17 [55] MOTA result. Furthermore, the proposed algorithm is capable of yielding solutions, which could be the start to a new pathway in the field of object tracking.

## 5. Limitations of the Study

This paper addressed only four semantic attributes, which could increase in future work by relabeling the shirts and trousers to include more clothes types. For instance, trousers can be relabelled to ‘jeans, shorts, leggings, or sweatpants’, and shirts can be relabelled to ‘blouse, T-shirt, crop top, blazer, jacket, or hoodie’. Thus, pedestrian attribute detection algorithms would benefit from our dataset as much as object detection algorithms. Moreover, it could be a start to fashion-based detection. Additionally, MSA-AF did not give the same performance in videos with zoomed-out scenes, where people’s appearance is rarely informative. This could be handled by adding more images to our dataset with a zoomed-out scene. Finally, as our algorithm is a tracking-by-detection algorithm, tracking performance relies on the detector’s output. Although our dataset increased the overall detector performance, failing to detect an object is still an issue. When the detector fails to detect an object, the tracker fails to track it, which increases false negatives. On the other hand, the five algorithms we compared our results with use a publicly provided detection set from MotChallenge as their input. This caused low MOTA and IDF1 in our algorithm compared to their algorithms.

## 6. What Went Wrong

After training scaled YOLOv4 on our dataset, we found some confusion between the shirts and the man and woman class. In other words, many man-classed objects were detected as man and shirt simultaneously, which confused our algorithm. When we looked back to the PGC dataset annotation, in some cases such as occlusion or a person raising their hand, the shirt Bbox included faces, confusing the detection model and making it think that any person could be shirt too (see Appendix A). We went back to the 40,000 images in our dataset to handle this problem and resized the shirt Bbox, so that no faces were included inside it. After that, we retrained scaled YOLOv4, which solved the problem.

## 7. Conclusions

This paper has presented a novel MOT algorithm that imitates the human brain in eight different areas, and a new dataset of 40,000 images and more than 190,000 annotations to serve the computer vision community. Opening the doors to more human brain-inspired algorithms that would change the way in which the tracking problem has previously been solved is one of the goals of this paper. In addition, there are encouraging directions for future annotations on the PGC dataset that would benefit a wide range of researchers. Finally, we presented our results on challenging videos from the MOTChallenge benchmark, where we measured the quantitative performance of MSA-AF compared to five of the most recent state-of-the-art trackers. These results show our algorithm’s impact on creating a stable tracker by decreasing the ML metric and increasing the MOTP at the same time. Moreover, our dataset gained 0.89 mAP compared to the 0.49 mAP from open image dataset v6.

## Figures and Tables

**Figure 1 sensors-21-07604-f001:**
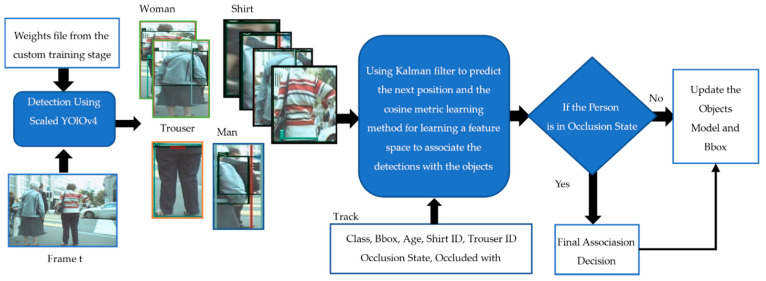
Overview of MSA-AF starting from the detection step, which feeds into the first association step. The final association step will be used in cases of occlusion. Track 1 to Track n represents the model of the object that has been tracked in the previous frames. A class could be 0, 1, 2, or 3, representing man, woman, shirt, and trousers. Bbox has x,y,h,w of the object’s bounding box, in which (x,y) represents the centre point, h is the height, and w is the width. The occlusion state is a binary number: ‘1’ if the object is in an occlusion state and ‘0’ if not. Occluded with has the ID of the object that was occluded. Age is the number of frames in which the object is successfully tracked. Finally, Semantic information includes much information that will be described in detail in Section 3. The original image is taken from the MOTChalleng dataset [28].

**Figure 2 sensors-21-07604-f002:**
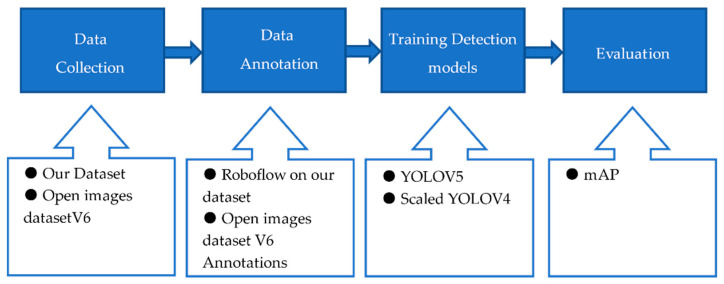
Steps that have been used to assert a high-performance detection model. Each phase included all options that we used for the comparison in order to get the best detection results.

**Figure 3 sensors-21-07604-f003:**
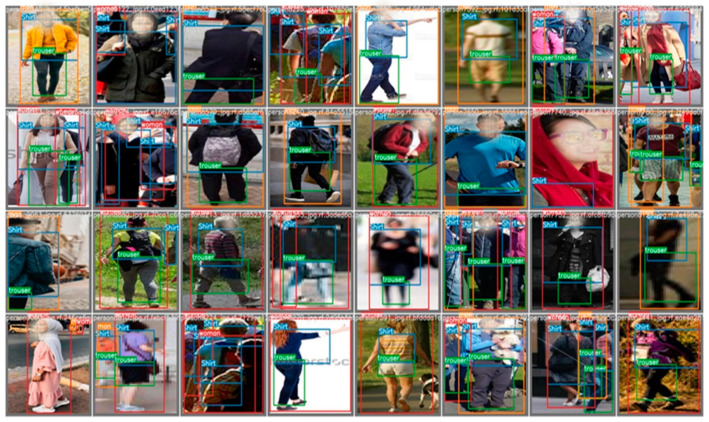
Images sample taken randomly from our dataset by Scaled Yolov4 to use in the training step; we did not choose any of them. These images prove the diversity of the PGC dataset, and it contains women, men, Arab women, Asian women, Black men, crowded scene, a single person scene, bluer scene, indoor scene, outdoor scene, back view, and front view.

**Figure 4 sensors-21-07604-f004:**
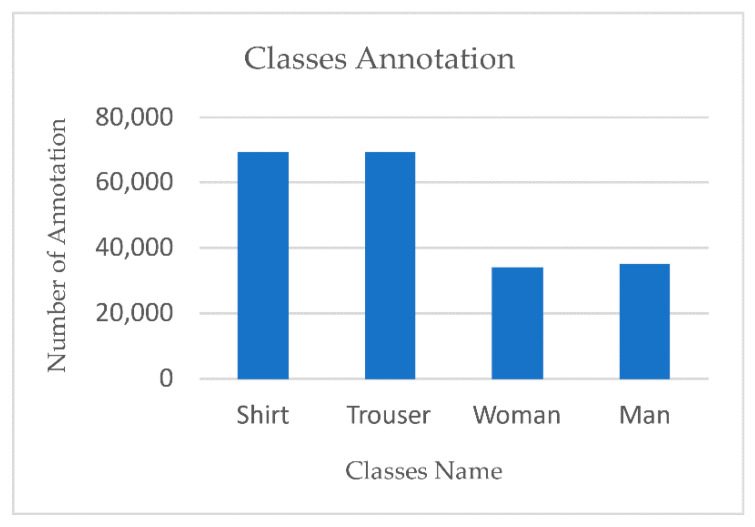
PGC class annotation balance.

**Figure 5 sensors-21-07604-f005:**
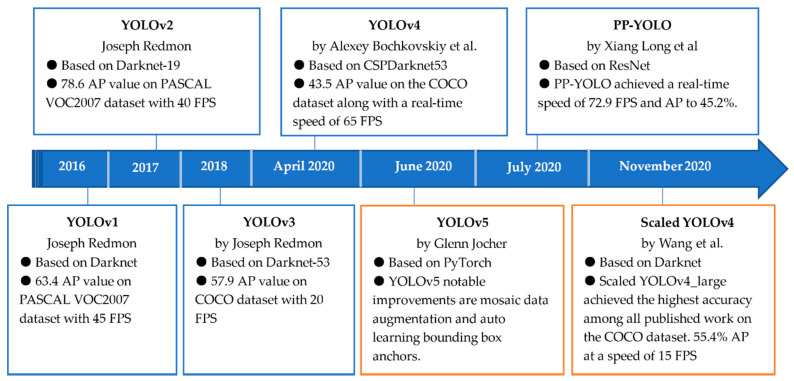
Timeline and performance comparison between members of the YOLO family.

**Figure 6 sensors-21-07604-f006:**
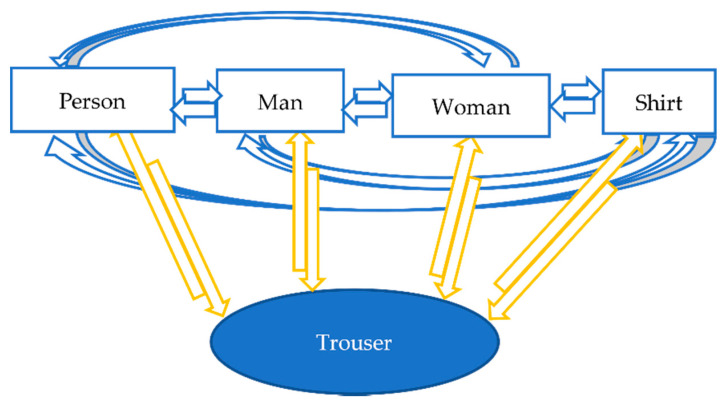
A combination of distributed-only and semantic hub for controlling the relationship between attributes.

**Figure 7 sensors-21-07604-f007:**
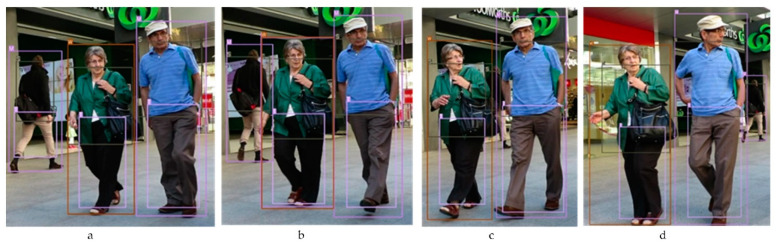
Four images taken from our detection outputs to demonstrate the occlusion phases. The original image is taken from the MOTChalleng dataset [28]. Before occlusion state (**a**), in occlusion state (**b**,**c**), and after occlusion state (**d**).

**Figure 8 sensors-21-07604-f008:**
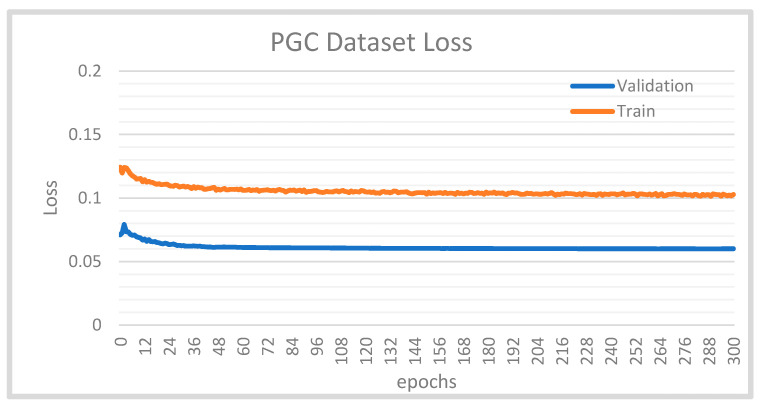
Our dataset loss plots for the training and validation sets.

**Figure 9 sensors-21-07604-f009:**
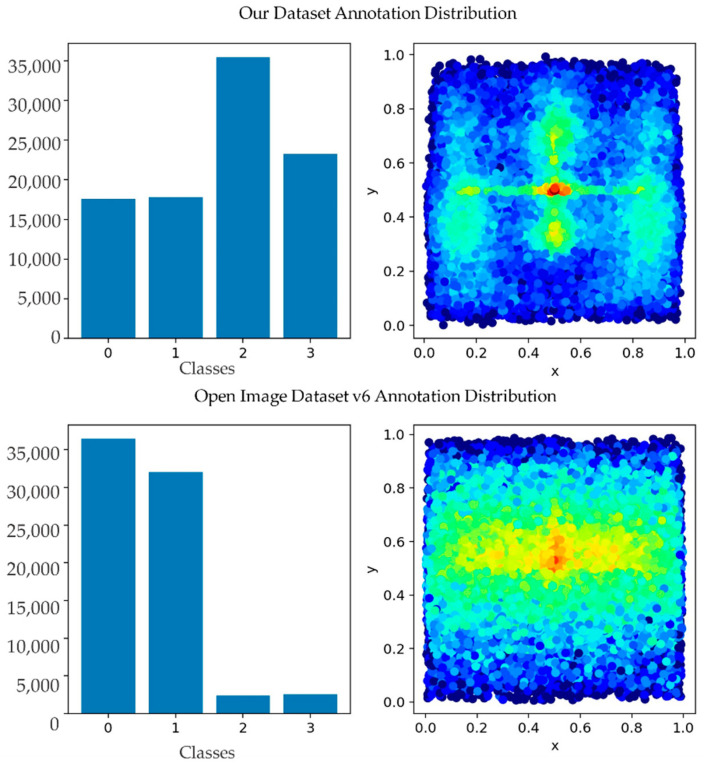
Comparison of class representation between the PGC dataset and open v6.

**Figure 10 sensors-21-07604-f010:**
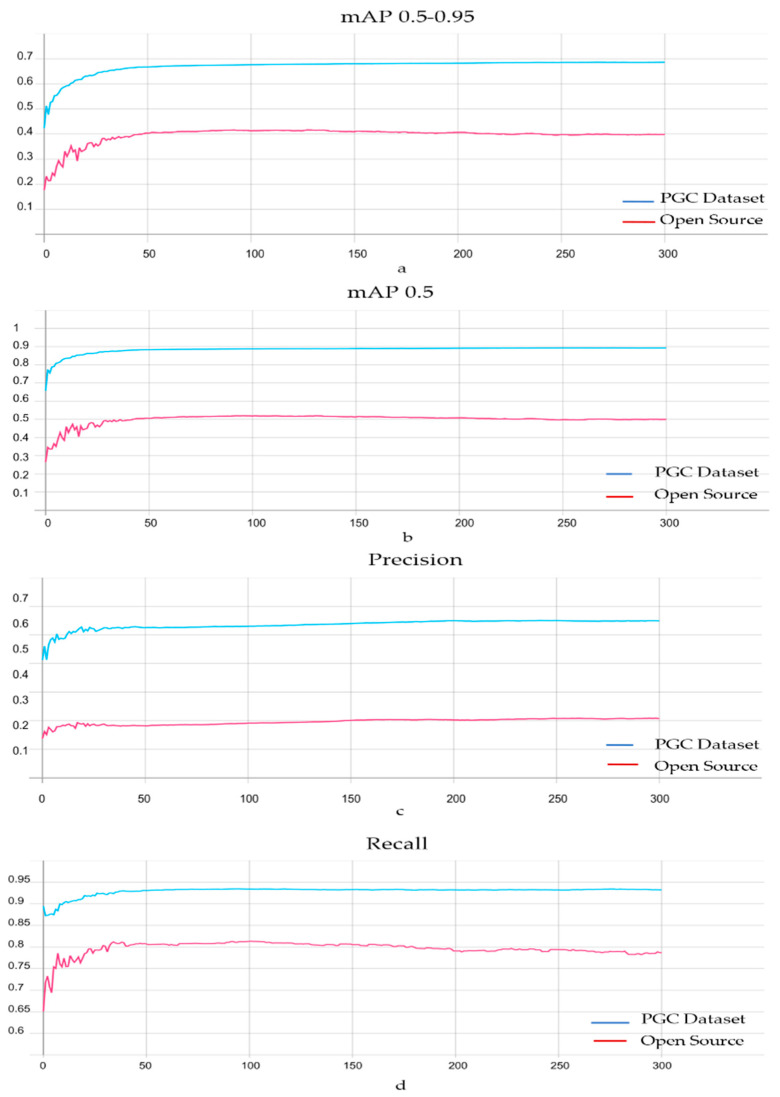
Four evaluation metrics, mAP 0.5–0.95 (**a**), mAP 0.5 (**b**), Recall (**c**), and Precision (**d**), that have been used to measure the performance of the PGC dataset and open dataset v6, after being used to train scaled YOLOv4. TensorBoard was used to create these graphics.

**Figure 11 sensors-21-07604-f011:**
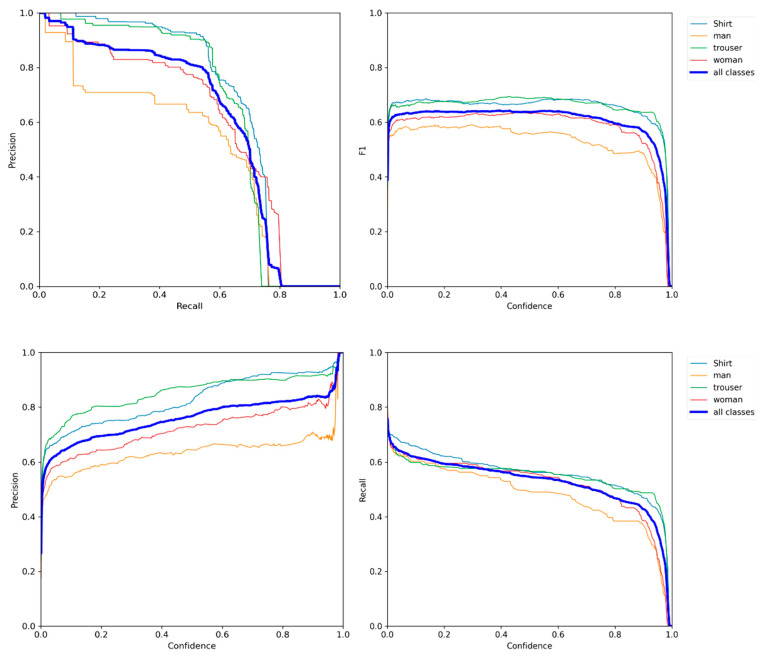
The performance metrics Precision, Recall, f1, and Precision-Recall of the four classes when the PGC dataset had been used to train YOLOv5. The YOLOv5 model was used to create these graphics.

**Figure 12 sensors-21-07604-f012:**
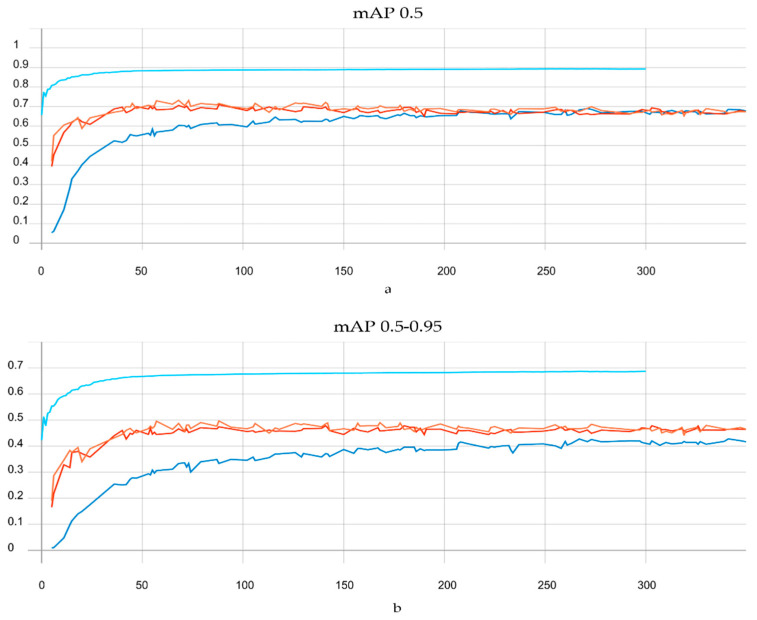
Performance of the four models, Scaled YOLOv4, YOLOv5l, YOLOv5s, and YOLOv5m using two metrics, mAP 0.5 (**a**) and mAP 0.5–0.95 (**b**). TensorBoard was used to create these graphics.

**Figure 13 sensors-21-07604-f013:**
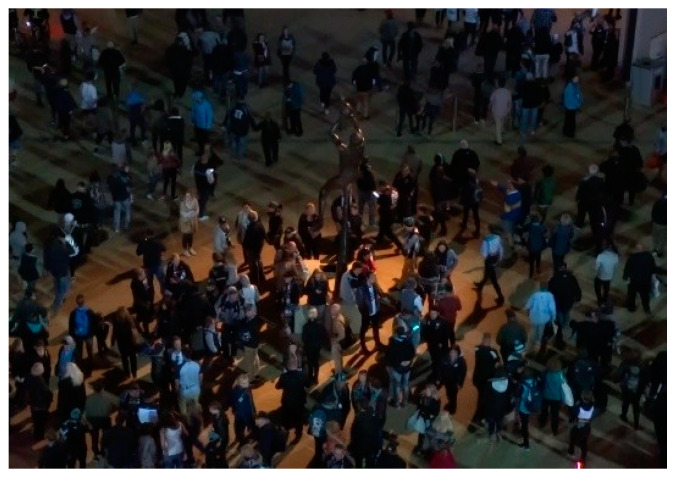
A sample from the MOT20-04 video [28].

**Table 1 sensors-21-07604-t001:** Human brain-inspired algorithms.

Type of Brain Inspiration	Contributions	Advantages and Limitations
Visual attention	Gao et al. [13]	Most attention-based algorithms can handle variability in background and object appearance. However, this adaptation makes these algorithms suffer from occlusion.
	Zhu et al. [14]	
	Chen et al. [15]	
Visual memory	Qi et al. [18]	Memory-based algorithms can successfully handle occlusion and data association, but they need to formulate vital parameters to decide what to remember and what to forget.
	Wang et al. [19]	
	Jiang et al. [20]	
	Kim et al. [21]	
	Wang et al. [22]	

**Table 2 sensors-21-07604-t002:** Eight human brain strategies behind our algorithm.

	Eight Human Brain Strategies to Handle Object Tracking	How MSA-AF Simulated These Strategies
1	Tracking in the human brain is a two-stage process; the first stage is for location processing, while the second stage is for identity processing [2].	MSA-AF is a tracking-by-detection algorithm. The first stage is detection (location process), then association (identifying process).
2	Experimental results by [23] suggest that the human brain uses motion prediction to handle the tracking problem.	MSA-AF used Kalman to predict the next position of the object.
3	Neural representation is needed to achieve particular goals in scenes, e.g., recognition [1].	MSA-AF used pre-trained CNN to compute the Bbox appearance descriptor in [24].
4	Neural representations reflect stimulus attributes in low-level visual areas and perceptual outcomes in high-level visual areas [4].	MSA-AF used semantic attributes (man, woman, shirt, trouser).
5	Experimental results by [17] conclude that the observer’s recovery from any errors during tracking was by storing surface properties in the visual working memory.	MSA-AF used the long-memory theory to save all information about each target and retrieve it when it is more needed, usually when the object is in an occlusion state.
6	The brain provides more attentional resources when objects are in a crowded scene, with a higher chance of being lost when target switching conditions happen [25].	Using (4), MSA-AF can decide if any object is in an occlusion situation or not. If yes, MSA-AF gives more attention to this object by using semantic information.
7	The human brain uses optimised features to discriminate targets better and retrieve them faster and more efficiently [2].	MSA-AF uses appearance information to discriminate the object.
8	When the possibility of confusing targets increases, it is suggested that human subjects benefit from rescue saccades (saccades toward targets that are in a critical situation) to avoid incorrect associations [26,27].	Final association decision at MSA-AF, Section 3 imitates this concept.

**Table 3 sensors-21-07604-t003:** Dataset Performance comparison with open image datasetv6 on scaled yolov4.

	Training Time	mAP 0.5	mAP 0.5–0.95	Precision	Recall
Open Images Dataset V6	3 d 5 h 11 min	0.4999	0.3983	0.2079	0.7867
PGC Dataset (ours)	**3 d 1 h 27 min**	**0.8924**	**0.6862**	**0.5487**	**0.9321**

**Table 4 sensors-21-07604-t004:** Performance comparison with five state-of-the-art algorithms on MOT17.

MOT17	↑ MOTA	↑ IDF1	↑ MOTP	↑ MT	↓ ML	↑ Rcll	↑ Prcn	↓ FAF
**LPC_MOT** [51]	**59**	**66.8**	78	**29.9**	33.9	63.3	93.9	1.3
**MPNTrack** [52]	58.8	61.7	78.6	28.8	33.5	62.1	95.3	1
**GNNMatch** [53]	57.3	56.3	78.6	24.4	33.4	60.1	96	0.8
**Tracktor++v2** [54]	53.5	52.3	78	19.5	36.6	56	**96.3**	**0.7**
**SORT17** [55]	43.1	39.8	77.8	12.5	42.3	49.0	90.7	1.6
**MSA-AF (Ours)**	44.4	37.1	**80.1**	13	**30.25**	50.6	89.55	**0.7**

**Table 5 sensors-21-07604-t005:** Performance comparison with five state-of-the-art algorithms on MOT20.

MOT20	↑ MOTA	↑ IDF1	↑ MOTP	↑ MT	↓ ML	↑ Rcll	↑ Prcn	↓ FAF
**LPC_MOT** [51]	56.3	**62.5**	79.7	34.1	25.2	**58.8**	96.3	2.6
**MPNTrack** [52]	**57.6**	59.1	79	38.2	22.5	61.1	94.9	3.8
**GNNMatch** [53]	54.5	49	79.4	32.8	25.5	56.8	96.9	2.1
**Tracktor++v2** [54]	52.6	52.7	**79.9**	29.4	26.7	54.3	**97.6**	**1.5**
**SORT17 [55]**	42.7	45.1	78.5	16.7	26.2	48.8	90.2	6.1
**MSA**-**AF (Ours)**	45.3	36.4	77.6	**40**	**19**	54.8	86.8	4.36

## Data Availability

Not applicable.

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
