# Peer review of "Brain Strategy Algorithm for Multiple Object Tracking Based on Merging Semantic Attributes and Appearance Features"

_sensors, 2021, doi:10.3390/s21227604_

Round 1

Reviewer 1 Report

1. There may be some errors in writing. For example, the IOD in line 585 of the paper is defined as IOA in line 546 of the paper? In addition, in lines 224 and 225, this should be a sentence? but be divided into different paragraphs. This problem also exists in lines 234 and 235.  In other places, the author is expected to do double-check and examination.
2. The paper declares that the semantic information of the tracking target without matching will be kept in memory.  With the growth of time, this information will be accumulated, how to better solve it?
3. The paper directly uses IOA for occlusion judgment, if it is greater than 0.9, it is considered that there is no occlusion, is this reasonable? For example, we use trousers to make a judgment. If another person's trousers completely enter the current person's box and block the current person's own trousers, because the trousers of different persons are not distinguished. Such a person is occluded, but according to the proposed algorithm, the person is not considered occluded.
4. The comprehensive indicators MOTA  and IDF1 indicators on the MOT17 and MOT20 datasets are too low compared with other methods. Has the author analyzed the reasons?  In addition, has the author done an experiment in the same environment, the ID of other methods is changed due to occlusion, but the method does not change. If so, please add it to the paper.

Reviewer 2 Report

  1. The quality of the figures did not reach a professional level. Resolution is not good. There is no unity on fonts, shapes, arrows, etc.
  2. The contents described in Table 2 are not comfortable to follow.
  3. Please explain logical evidences or practical effects for the readers to accept that the introduced new dataset can be a contribution.
  4. How can two datasets can be evaluated and compared just by using the same detection model. Can you prove there is no bias or over-fitted factors in the new dataset?
  5. How can we say that the proposed MSA-AF is competitive. The tables show pros and cons at the same time.
  6. The authors said that there are four contributions but only three have been mentioned.

Round 2

Reviewer 1 Report

It can be accepted in the present form.

Author Response

Thank you for the respected viewer.

Reviewer 2 Report

The authors have kindly addressed the reviewer's comments.

However, the figures have not been meet the minimum professional level required by a scientific journal. The reviewer suggest the authors reconsider whether the figures in the manuscript are of reasonable quality compared to the figures found in books in bookstores. 

  1. All figures should be processed as vector type. Every pixel except cropped real image should be maintained without distortion no matter how much the manuscript pdf is enlarged.
  2. Figure 1 is unnecessarily large, and the font size in the figure is also a bit large compared to the plain text font.
  3. The shape and alignment of Figure 2 are still not professional.
  4. Figure 4, fonts,,, please use another graph generation tool and extract the figure as pdf or eps format. 
  5. The shape and alignment of Figure 5~11 (there is also a typo in figure number and caption) are still not professional. They look like they have been captured from another source. In other word, they look like not from the authors' raw data. 

Author Response

We want to thank the respected reviewer for the detailed comments that helped us improve the quality of the figures. We looked through all the figures and have changed nine figures in the manuscript. We will attach these Figures with the updated manuscript too.